# Mummified Wood of *Juniperus* (Cupressaceae) from the Late Miocene of Taman Peninsula, South Russia

**DOI:** 10.3390/plants11152050

**Published:** 2022-08-05

**Authors:** Anna V. Stepanova, Anastasia A. Odintsova, Alena I. Rybkina, Yuliana V. Rostovtseva, Alexei A. Oskolski

**Affiliations:** 1Komarov Botanical Institute of the Russian Academy of Science, Professor Popov Street 2, 197376 St. Petersburg, Russia; 2Geophysical Center of the Russian Academy of Sciences (GC RAS), Molodezhnaya Street 3, 119296 Moscow, Russia; 3Department of Petroleum Sedimentology and Marine Geology, Faculty of Geology, Lomonosov Moscow State University, Leninskie Gory GSP-1, 119991 Moscow, Russia; 4Department of Botany and Plant Biotechnology, University of Johannesburg, Auckland Park 2006, Johannesburg P.O. Box 524, South Africa

**Keywords:** Cupressaceae, Eastern Paratethys, Maeotian, wood anatomy, conifers

## Abstract

*Juniperus* L. is the second-largest genus of conifers, having the widest distribution of all conifer genera. Its phytogeographic history is, however, obscure due to its very poor fossil record. We described a wood of *Juniperus* sp. from the lower Maeotian sediments of the Popov Kamen section, Taman Peninsula, South Russia, in order to clarify its taxonomic position shedding light on the phytogeographic history of the genus. This fossil wood was well-preserved by mummification, which allowed for it to be studied by the same methods as used for the anatomical examination of modern woods. The wood from the Popov Kamen section shows the greatest similarity to the extant Mediterranean species *J. excelsa*, belonging to the section *Sabina*. This is the first reliable macrofossil evidence of the sect. *Sabina* from Eurasia convincingly dated to the Miocene. The age of the mummified wood from the Popov Kamen section is consistent with molecular dating of diversification of the lineage comprising juniper species of the sect. *Sabina* from Europe, Asia and eastern Africa. The wood of *Juniperus* sp. has not been buried in situ, as it was found in the relatively deep-water marine sediments. The available coeval pollen series and macrofossils of Cupressaceae from the surrounding regions suggest that this wood was likely transferred by sea current from the northwestern side of the Black Sea, which was a part of the Eastern Paratethys.

## 1. Introduction

*Juniperus* L. is the second-largest genus of conifers, and the largest member of the family Cupressaceae. Comprising 75 species [1], *Juniperus* has the widest distribution of all conifer genera [2]. Most species of this genus are confined to forests and to shrubby vegetations occurring in arid and semiarid regions throughout the Northern Hemisphere, with a single species crossing the Equator in eastern Africa [3]. Adams [1] recognized three monophyletic sections within this genus: sect. *Calocedrus* Endl., with a single species in the Mediterranean; sect. *Juniperus*, with fourteen species in East Asia and the Mediterranean plus one circumboreal species, *J. communis* L.; and sect. *Sabina* Spach, with 60 species distributed in southwestern North America, Asia and the Mediterranean as well as in eastern Africa and Macaronesia.

Molecular dating with fossil calibration shows that the *Juniperus* diverged from other Cupressaceae in the late Paleocene to Eocene [4,5]. Axelrod [6] suggested that diversification of junipers occurred within warm temperate semiarid vegetation of the Madrean–Tethyan belts that ran along the southern areas of Eurasia and North America during Eocene and Oligocene. This scenario has been confirmed by molecular evidence [4]. The fossil record for *Juniperus* is still too poor for comprehensive reconstructions of the history of this genus.

The most ancient fossils of *Juniperus* belonging to the sect. *Sabina* have been described from the Eocene–Oligocene boundary in the Czech Republic [7]. Other fossils attributed to this section were reported from the Oligocene to the middle Miocene deposits of North America [8,9,10,11], from the Miocene–Pliocene boundary of Bulgaria [12] and from the Pliocene of Bashkortostan, Russia [13]. At the same time, no reliable macrofossil evidence for diversification of the sect. *Sabina* during the Miocene has been found in Eurasia. As for the sections *Juniperus* and *Calocedrus*, their fossils are known only from the middle Miocene and the Pliocene of Europe [12,14,15,16].

In this study, we present anatomical investigations of mummified wood belonging to the genus *Juniperus* from the Late Miocene sediments of the Popov Kamen section, Taman Peninsula, South Russia. We aim to clarify the taxonomic position of this fossil wood in order to shed light on phytogeographical history of the section *Sabina* during the Neogene.

## 2. Results

### 2.1. Systematic Description

Order **Coniferales** Gorozhankin, 1904

Family **Cupressaceae** Gray, 1822

Genus ***Juniperus*** Linnaeus, 1753 [type: *Juniperus communis* L.]

Section ***Sabina*** Spach, 1841

***Juniperus*** sp.

Material: PK-2020, three fragments of well-preserved mummified wood from an entire fossil stem, discovered from the Late Miocene sediments of the Popov Kamen section, Taman Peninsula, South Russia, deposited at the Laboratory of Paleobotany, Komarov Botanical Institute, St. Petersburg, Russia. Duplicates of these samples were deposited at the Department of Paleontology, Geological Faculty, Moscow State University, Moscow, Russia (Figure 1).

### 2.2. Description

Growth rings are distinct, 0.19–1.60 mm wide; the transition from earlywood to latewood is gradual (Figure 2A). Earlywood tracheids are thin-walled (1.8–3.6 μm thick), polygonal to oval in outline, 13–31 μm (mean 22 μm) in tangential diameter. Latewood tracheids are thin- to moderately thick-walled (2.1–3.8 μm thick), circular to oval in outline, 12–24 μm (mean 11 μm) in tangential diameter. False growth rings occur. Normal and traumatic axial resin ducts are not found. The bordered pits on the radial tracheid walls (Figure 2F) are uniseriate, circular to oval in outline, with 5–9 μm in diameter and prominent tori. Small bordered pits (3–5 μm) also occur on the tangential walls of the tracheids. Warty layer, crassulae, helical and callitroid thickenings on the tracheid walls are not found.

The axial parenchyma is abundant, tangentially zonate (Figure 2A) and occasionally also marginal (Figure 2B); in strands of 2–4 cells with thickened pitted longitudinal walls and nodular transverse walls (Figure 2C), sometimes with distinct indentures. Dark deposits commonly occur in axial parenchyma cells (Figure 2C).

Rays are exclusively uniseriate (Figure 2D), completely composed of parenchyma cells (ray tracheids absent), 1–9 cells high (mean 3.8 cells); ray cells are 17–28 μm (mean 22.7 μm) in height. The horizontal walls of ray parenchyma cells are moderately thick and distinctly pitted (Figure 2E); the end walls of ray parenchyma cells are nodular, with distinct indentures. Cross-field pits are cupressoid (Figure 2F) and taxodioid, circular to oval, 4–6 μm in diameter of; cross-fields mostly have 2–4 pits (up to 5) pits on marginal portions of rays, and mostly 1–2 (up to 4) pits in central regions of rays. Radial resin ducts are not found.

Dark-stained compounds are common in axial parenchyma cells and occur in ray cells. Crystals are not found.

### 2.3. Comparison with Modern Woods

The fossil sample from Taman Peninsula represents a typical homoxylic wood, showing the tracheids bearing large (up to 24 µm in diameter) circular bordered pits on radial walls as well as exclusively uniseriate rays. This suite of traits occurs only in conifers. The InsideWood [17] search for the combination of distinct growth-ring boundaries (40p), absence of helical thickenings on tracheid walls (61a), tangentially zonate axial parenchyma (72p, 74p), absence of ray tracheids (80p), distinctly pitted end walls of ray parenchyma cells (86p), cupressoid cross-field pits (93p), average ray height < 4 cells (102p, 103a, 104a, 105a), absence of axial canals (109a), radial canals (110a) and traumatic canals (111a) returns *Fitzroya cupressoides* (Molina) I.M. Johnst. and eight speices of *Juniperus* (*J. californica* Carrière, *J. chinensis* L., *J. drupacea* Labill. *J. excelsa* M.Bieb., *J. oxycedrus* L., *J. phoenicia* L., *J. rigida* Siebold & Zucc., *J. squamata* Buch.-Ham. ex D.Don.). Among these species, the presence of marginal axial parenchyma has been reported only in *J. drupacea* [18], *J. excelsa* and *J. oxycedrus* [19,20]. *J. drupacea* and *J. oxycedrus* are distinctive, however, from the fossil wood from Taman Peninsula by larger pits on tangential tracheid walls (6–10 µm and 8–10 µm in diameter, respectively, vs. 3–5 µm in the sample under study) and also by the occurrence of higher (>10 cells in height, up to 18 cells in *J. drupacea*) rays. The latter species also differs from the wood sample under study by fewer (mostly 1–2) pits per cross-fields [21]. *J. excelsa* shows greater similarity with the Miocene wood, but it differs from the fossil wood in thinner horizontal walls of the ray cells, and the occasional occurrence of biseriate rays [19,20,21].

As the presence of marginal axial parenchyma is a prominent feature of the fossil sample from Taman Peninsula, which is uncommon in the wood of conifers, we compared it with other extant *Juniperus* species having this trait. Apart from *J. drupacea*, *J. excelsa* and *J. oxycedrus*, marginal axial parenchyma has been reported in *J. conferta* [18], *J. monosperma* (Engelm.) Sarg., *J. thurifera* L., *J. scopulorum* Sarg. and *J. tibetica* Kom. [19,20]. *J. scopulorum* and *J. thurifera* differ from the fossil wood from Taman Peninsula by the occurrence of larger pits (>10 µm in diameter) on tangential walls of tracheids [21]. *J. thurifera* is also distinctive by much more numerous rays (190–200 rays per mm^2^ vs. 93–117 rays per mm^2^, respectively), lower ray cells (10–18 µm vs. 17–28 µm in height, respectively), and the lack of indentures [19,21]. Unlike our fossil wood, *J. conferta* has mostly unpitted end walls of ray cells [18,21]. Then, *J. scopulorum*, *J. monosperma*, and *J. tibetica* share the occurrence of higher rays (>10 cells in height, up to 16 cells in *J. scopulorum*) than the fossil wood from Taman Peninsula. The latter two species and *J. conferta* are also distinctive from the sample under study in having fewer (up to three) pits on cross-fields [21].

In summary, our fossil wood belongs to the genus *Juniperus*, but it cannot be convincingly placed into any modern species that have been examined to date by wood anatomists. It shows the greatest similarity to the Mediterranean species *J. excelsa*, belonging to the section *Sabina*.

### 2.4. Comparison with Fossil Woods

The fossil homoxylic woods showing a combination of spaced arrangement of circular pits on radial tracheid walls, cupressoid cross-field pits, nodular transverse walls of axial parenchyma cells and usually also nodular walls of ray cells have been ascribed to the genera *Juniperoxylon* Houlbert [22,23] and *Juniperus* [20,24]. *Juniperoxylon pottoniense* (Stopes) Kräusel from the early Cretaceous of England [25] and the Eocene of Denmark [26] and *J. wagneri* Süss & Rathner from the Miocene of Germany [27] are distinctive, however, from other congeneric species and from the fossil wood under study in having smooth horizontal walls of ray cells. The latter species as well as other members of *Juniperoxylon*, including *J. zamunerae* Ruiz & Bodnar from the Middle Triassic of Argentina [23,28]; *J. breviparenchmatosum* Watari & Nishida from the Eocene of Hokkaido, Japan [29]; *Juniperoxylon acarcae* Akkemik, from the early Miocene of the central Turkey [20]; and four species from the Miocene deposits of Germany *J. juniperoides* (Kownas) Huard, *J. pachyderma* (Göppert) Kräusel, *J. rhenanum* Burgh from North Rhine-Westphalia [30] and *J. schneiderianum* Dolezych from Lusatia [31], differ from the fossil wood from the Taman Peninsula by the presence of biseriate pitting on the radial tracheid walls, and by the occurrence of biseriate rays. Finally, two fossil woods of *Juniperus* sp. from the early Miocene of the Galatean Volcanic Province, northwestern Turkey [24,32], show greatest resemblance to the studied wood sample, but both are distinctive in the lack of marginal axial parenchyma.

Overall, the mummified wood from the late Miocene deposits of the Taman Peninsula shows a close affinity to some extant species from the section *Sabina* of the genus *Juniperus*, as well as to the early Miocene woods from the northwestern Turkey assigned to the *Juniperus* sp. Although the studied sample is distinctive from any woods of modern or extinct junipers described to date, its anatomical traits are not sufficient for its reliable taxonomic attribution. Thus, we do not consider the fossil wood as a new species of this genus, but designate it as *Juniperus* sp. seemingly belonging to the section *Sabina*.

## 3. Discussion

The mummified wood of *Juniperus* sp. from the lower Maeotian sediments of Taman Peninsula is the first reliable macrofossil evidence of the section *Sabina* from Eurasia whose age is convincingly dated to the Miocene. The most ancient fossils of junipers attributed to this group have been described from the Eocene/Oligocene boundary of north Bohemia, the Czech Republic [7]. More recent Neogene record of the section *Sabina* includes three extinct species from the Oligocene to the middle Miocene of the North America [8,9,10,11] as well as the fossil twigs ascribed to the extant *J. foetidissima* from the Miocene–Pliocene boundary from the Sofia Basin, Bulgaria [12], and the seeds of *J. sabina* from Bashkortostan, Russia [13]. The mummified wood of *Juniperus* sp. from the Popov Kamen section shows that the species of this lineage occurred in the regions adjacent to Eastern Paratethys at least since the early Maeotian age.

Obviously, the wood of *Juniperus* sp. has not been buried in situ, as it was found in relatively deep-water marine sediments. The only coeval occurrence of the pollen grains ascribed to *Juniperus* sp. has been reported from the lower Maeotian deposits of Odessa Oblast, southern Ukraine [33]. More ancient pollen evidence for this genus was found in the Sarmatian deposits of Kartli, eastern Georgia [34]. As for the macrofossils, the cone of *J. bessarabica* Negru has been described from the lower Sarmatian of Moldova [14] (Figure 3). This extinct species shows greatest affinity to the section *Juniperus*, i.e., to another lineage of junipers than the fossil wood under study attributed to the section *Sabina*. Cupressaceae have not been reported, however, in other pollen series studied in the Sarmatian and Maeotian deposits of the Eastern Paratethys regions, including those from the Taman Peninsula [35,36], the Lower Don [37], Bulgaria [38], Abkhazia [39] and several localities of Georgia [40]. No macrofossils of *Juniperus* have also been found in coeval paleofloras in southern Ukraine [33], Krasnodar Krai of Russia [41] and Georgia [40,42,43]. The available data suggest, therefore, that the wood of *Juniperus* sp. was likely transferred by sea current from the northwestern side of the Black Sea, which was a part of the Eastern Paratethys in the Miocene. Transportability of driftwoods over large distances has been supported by strong evidence [44].

The fossil wood of *Juniperus* sp. shows the greatest similarity to the extant Mediterranean species *J. excelsa*, belonging to a well-supported “clade IV” within the section *Sabina* [4]. This lineage also comprises the European species *J. thurifera*, the Asian *J. chinensis* and *J. polycarpus*, as well as *J. procera* from east Africa and south Arabia. As suggested by molecular dating [4], this lineage has been diversified during the Miocene. This estimation is consistent with the age of the fossil wood of *Juniperus* sp. from the Popov Kamen section. The reported fossil records of this group, up to now, are too sparse, however, for detailed reconstruction of its phytogeographic history.

**Figure 3 plants-11-02050-f003:**
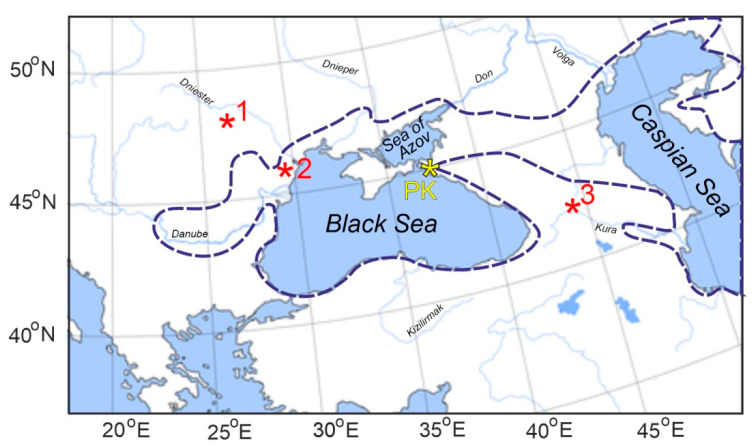
Locations of the Popov Kamen section (PK, yellow asterisk) and coeval fossils of *Juniperus* from the regions adjacent to the Black Sea (red asterisks). The Paratethys area configuration at the Late Miocene is marked by dotted line. 1: The cone of *Juniperus bessarabica* Negru from the lower Sarmatian of Moldova [14]. 2: Pollen of *Juniperus* sp. from the lower Maeotian of southern Ukraine [33]. 3: Pollen of *Juniperus* sp. from the Sarmatian of eastern Georgia [34].

## 4. Materials and Methods

Three fragments of totally mummified portion of entire tree trunk, 23 cm in length and 7 cm in diameter, without any traces of organisms feeding (Figure 1), were collected from the Upper Miocene sediments of the Popov Kamen section (Figure 3 and Figure 4A,B). This trunk shows about 50 growth rings (some rings are hardly detectable) on its cross section.

Since its first description by Andrusov [45], this geological section has been extensively studied, using paleomagnetic, paleontological, and lithological methods of investigation [45,46,47,48,49,50,51,52]. The Popov Kamen section is located on the Black Sea coast of Taman Peninsula (45°16′01.8″ N, 36°61′97.6″ E, Russia) and comprises well-exposed Upper Sarmatian as well as Lower and Upper Maeotian sediments of the Eastern Paratethys. These sediments mainly represent clays with sporadic diatomite and limestone layers. The large bryozoan build-ups are located at the base of the Lower Maeotian, which directly underlie and overlie clays. The studied fossil wood was found in the clays laying 1.5–2 m above the top of the large bryozoan build-ups; these clays contain no fauna. The Lower Maeotian sediments of the Popov Kamen section accumulated in relatively deep-water environments (at depths of 50–75 m [46,47]). The Maeotian started with a marine transgression, which increased salinity back to 18‰ and flooded marginal areas of the Eastern Paratethys. At the Late Sarmatian, the Eastern Paratethys was mainly isolated. The sea level became unstable and a regression caused exposure of marginal parts of the basin, as well as a significant decrease in salinity down to 4–9‰ [53].

**Figure 4 plants-11-02050-f004:**
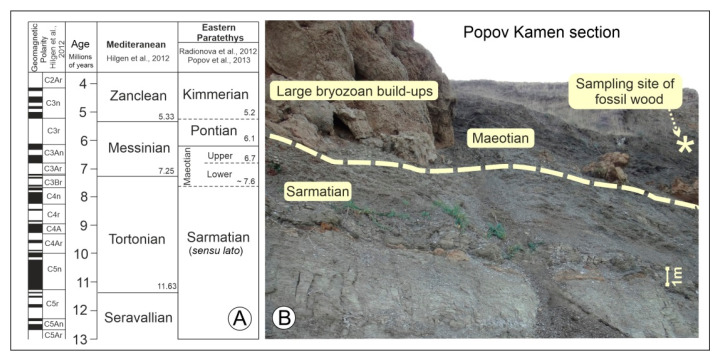
Geological setting of the mummified wood’s locality. (**A**) The time scale for Mediterranean and Eastern Paratethys (Black Sea). The ages of boundaries between the stages are indicated (millions of years). (**B**) The location of fossil woods; the boundary between the Sarmatian and Maeotian sediments was marked by yellow hashed line [50,51,52,54,55].

The fossil wood in this study was well-preserved by mummification. This specimen was processed and sectioned using the same methods as used for modern wood. The wood samples were boiled in water for about one hour and sectioned with a sledge microtome. Transverse, radial and tangential microtome sections of 20–30 µm in thickness were stained with alcian blue/safranin [56] or left unstained, dehydrated in gradient series of alcohol, and then mounted in Euparal. Then, the sections were examined with a light microscope (Olympus BX53). Wood anatomical measurements and anatomical terminology used for the descriptions in this paper follow the recommendations of the International Association of Wood Anatomists (IAWA) list of Microscopic Features for Softwood Identification [56]. The taxonomic position of fossil woods is determined by comparative work with similar modern and fossil softwood structures. This comparative work is based on modern wood slices and reference materials, particularly the computerized InsideWood database [57], in which we can search for similar modern softwoods.

## Figures and Tables

**Figure 1 plants-11-02050-f001:**
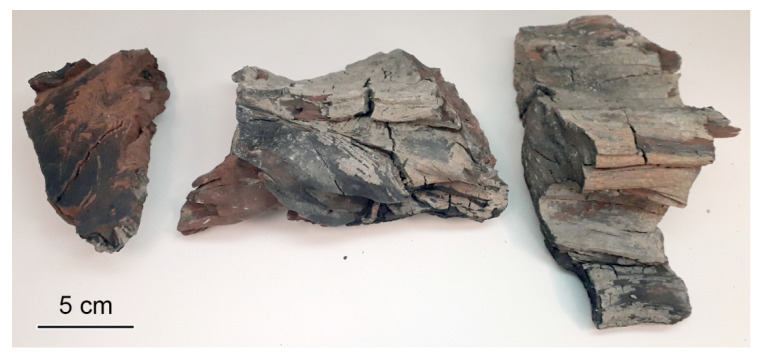
Fragments of fossil woody stem examined in the present study.

**Figure 2 plants-11-02050-f002:**
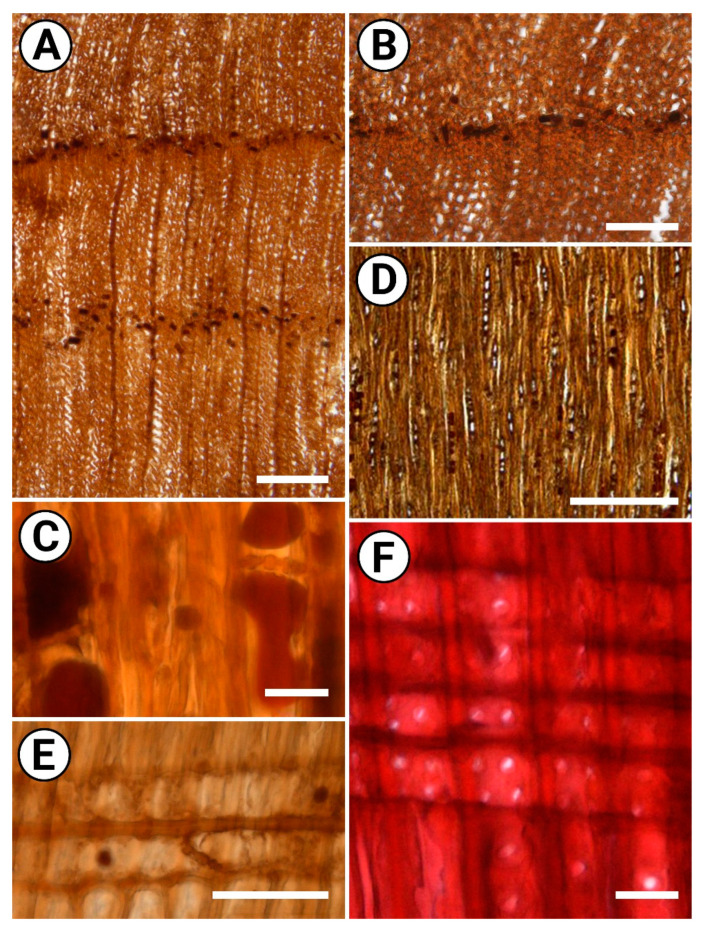
Wood structure of *Juniperus* sp., light microscopy. (**A**) Transverse section (TS), distinct growth-ring boundary, gradual transition from earlywood to latewood, tangentially zonate axial parenchyma. (**B**) TS, distinct boundary of growth ring, marginal axial parenchyma. (**C**) Tangential longitudinal section (TLS), portions of axial parenchyma strands, nodular transverse wall, dark deposits in axial parenchyma cells. (**D**) TLS, exclusively uniseriate low rays (up to 8 cells in height). (**E**) Radial longitudinal section (RLS), ray cells with thickened pitted horizontal walls and nodular end walls. (**F**) RLS, bordered pits on radial tracheid walls, cross-fields with 2–4 cupressoid pits. Scale bars: 200 µm for (**A**,**D**), 100 µm for (**B**), 50 µm for (**C**,**E**), 20 µm for (**F**).

## Data Availability

Not applicable.

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
