# Peer review of "Mummified Wood of *Juniperus* (Cupressaceae) from the Late Miocene of Taman Peninsula, South Russia"

_plants, 2022, doi:10.3390/plants11152050_

Round 1
Reviewer 1 Report
Introduction
The manuscript Mummified wood of Juniperus (Cupressaceae) from the Late Miocene of Taman Peninsula, South Russia submitted for review is well written and contains valuable information about the finding. The description provides comprehensive information on the identification features of the studied wood fragment.
Critique
The paper lacks a clearly formulated purpose, so it is difficult to determine whether the authors achieved the effect they intended. Was the purpose of the paper to describe the anatomical features of a piece of wood? Was it to determine the botanical affiliation? Was it to determine the origin of the wood in the find? etc. Since the work was submitted to the “Plant Systematix, Taxonomy, Nomenclature and Classification” section, it rather relates to this topic. The purpose of the work should be clearly stated, both in the text and the abstract.
The research methods are well described. The only thing missing is information about the mummified wood. Was the piece a complete piece of trunk or branch from the point of view of cross-section (from pith to bark)? If so, was the number of annual rings determined? This could be important information for possible dendrochronological studies. If there was only an incomplete piece also this should be written.
Was the piece mummified totally or did it indicate traces of other processes like petrification, lignification?
Photos of microscope slides are rather poor technical quality, but acceptable, if it is possible I recommend provide better quality (especially 1C, 1E and 1F). The photo in Figure 3B also show low quality.
What do the symbols in Figure 3 mean? ‘Ma’ in column 2, s.l. (under Sarmatian), numbers 5.33, 7.25 … etc.
Please check the bibliographic concordance of item 53 (ll. 345-346). It seems that the cited item was published by Elseviere and was titled ‘The Geological Time Scale 2012’ instead of ‘A Geological Time Scale 2012’.
Author Response
We are very grateful to the Reviewer the careful reading our manuscript and for the critical comments. We agree with the changes suggested by the Reviewer: the manuscript was really improved after these corrections.
Reviewer 1
The paper lacks a clearly formulated purpose, so it is difficult to determine whether the authors achieved the effect they intended. Was the purpose of the paper to describe the anatomical features of a piece of wood? Was it to determine the botanical affiliation? Was it to determine the origin of the wood in the find? etc. Since the work was submitted to the “Plant Systematix, Taxonomy, Nomenclature and Classification” section, it rather relates to this topic. The purpose of the work should be clearly stated, both in the text and the abstract.
We added the sentence in the last paragraph of the Introduction: “We aim to clarify the taxonomic position of this fossil wood, in order to shed light on phytogeographical history of the section Sabina during the Neogene.” Similar phrase was added in the Abstract.
The research methods are well described. The only thing missing is information about the mummified wood. Was the piece a complete piece of trunk or branch from the point of view of cross-section (from pith to bark)? If so, was the number of annual rings determined? This could be important information for possible dendrochronological studies. If there was only an incomplete piece also this should be written.
Was the piece mummified totally or did it indicate traces of other processes like petrification, lignification?
We corrected the first paragraph of the Materials and methods in the following way: The totally mummified portion of entire tree trunk of 10 cm in length and 5 cm in diameter was collected from the Upper Miocene sediments of the Popov Kamen section (Figure 2, 3A–B). This trunk shows about 50 growth rings (some rings are hardly detectable) on its cross section.
Photos of microscope slides are rather poor technical quality, but acceptable, if it is possible I recommend provide better quality (especially 1C, 1E and 1F). The photo in Figure 3B also show low quality.
We tried to improve these pictures, but, unfortunatelly, we were not able take the microphotos of better quality with the available microscope and digital camera.
What do the symbols in Figure 3 mean? ‘Ma’ in column 2, s.l. (under Sarmatian), numbers 5.33, 7.25 … etc.
“Ma” = “millions of years”, “s.l.” = sensu lato. We added these unabbreviated explications in the image.
“numbers 5.33, 7.25 …” are ages of boundaries between the sections. We added this explication (The ages of boundaries between the stages are indicated (millions of years)) in the caption of Fig. 3/
Please check the bibliographic concordance of item 53 (ll. 345-346). It seems that the cited item was published by Elseviere and was titled ‘The Geological Time Scale 2012’ instead of ‘A Geological Time Scale 2012’.
Done. Thanks!
Reviewer 2 Report
The paper presents anatomical investigations of mummified wood belonging to the genus Juniperus from the Late Miocene sediments of the Popov Kamen section, Taman Peninsula, South Russia. The methods employed for its characterization are the same methods as used for anatomical examination of modern woods. The paper is overall well-written, all resulted data are clearly presented and discussed. The references section is appropriate and related to the topic of the paper.
As a minor comment, please verify the text for editing errors, including the abstract.
Author Response
We are very grateful to the Reviewer for the careful reading our manuscript and for the critical comments. We agree with the changes suggested by the Reviewer: the manuscript was really improved after these corrections.
As a minor comment, please verify the text for editing errors, including the abstract.
We doublechecked the text, and made several minor corrections in the Abstract, Discussion and Materials and Methods.